# Scaling and context steer LLMs along the same computational path as the human brain

**Joséphine Raugel**
Meta AI
Laboratoire de Neurosciences Cognitives
et Computationnelles (Inserm U960)
Ecole Normale Supérieure - PSL

**Stéphane d'Ascoli**
Meta AI

**Jérémy Rapin**
Meta AI

**Valentin Wyart\***
Laboratoire de Neurosciences Cognitives
et Computationnelles (Inserm U960)
Ecole Normale Supérieure - PSL

**Jean-Rémi King\***
Meta AI

\*shared senior authorship

## Abstract

Recent studies suggest that the representations learned by large language models (LLMs) are partially aligned to those of the human brain. However, whether and why this alignment score arises from a similar sequence of computations remains elusive. In this study, we explore this question by examining temporally-resolved brain signals of participants listening to 10 hours of an audiobook. We study these neural dynamics jointly with a benchmark encompassing 17 LLMs varying in size and architecture type. Our analyses confirm that LLMs and the brain generate representations in a similar order: specifically, activations in the initial layers of LLMs tend to best align with early brain responses, while the deeper layers of LLMs tend to best align with later brain responses. This brain-LLM alignment is consistent across transformers and recurrent architectures. However, its emergence depends on both model size and context length. Overall, this study sheds light on the sequential nature of computations and the factors underlying the partial convergence between biological and artificial neural networks.

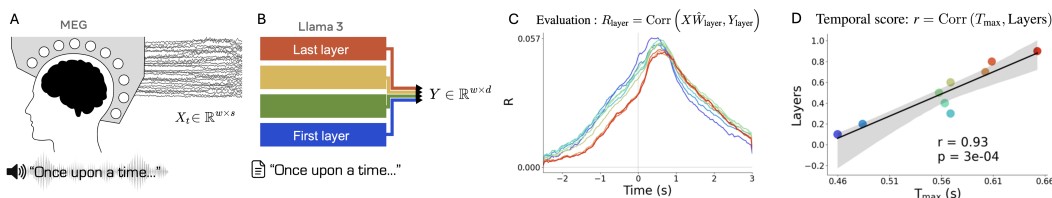

Figure 1: **Methods.** A. Subjects listened to 10 hours of audio books in the MEG scanner. B. The same text is input to an LLM, e.g. Llama 3-8B. Colors indicate layer depth. To compare this set of - biological and artificial - neural embeddings, we fit a linear mapping W for each layer, and evaluate its accuracy with a Pearson correlation metric: the alignment score $R_{layer}$. C. Alignment score ($R_{layer}$) of 9 representative layers of Llama 3-8B, as a function of word-onset (t=0). D. The timestep of peaking alignment scores ($T_{max}$, x-axis) is plotted for each layer (y-axis). The resulting Temporal score $r$ and associated $p$ are printed on the plot.

39th Conference on Neural Information Processing Systems (NeurIPS 2025).

# 1   Introduction

**Motivation.**   While large language models (LLMs) are not designed to resemble the human brain, recent studies show that their activations share similarities with those of the brain in response to speech [1, 2, 3, 4, 5, 6]. In the same way bats and birds independently evolved wings [7], LLMs and the human brain thus seem to follow a partial convergence [4].

**State of the art.**   Recent studies showed that an anatomical alignment exists between LLM layers and functional regions of the human brain, in the sense that their first layers tend to align with low-level areas of the brain such as primary sensory cortices whereas their deeper layers tend to align with higher-level areas such as secondary sensory or associative cortices [5, 8, 3, 9, 1].

**Remaining challenges.**   While this *anatomical* alignment between LLMs and the brain is increasingly established, the *order* in which these representations emerge remains poorly understood. While it has been shown by [6] that GPT2-XL exhibits a form of temporal alignment, (i) whether this temporal alignment with the brain is systematically found across LLMs, (ii) whether this alignment depends on the type of architecture, (iii) on its size, or (iv) on the length of its context remains currently unknown. In sum, the factors that lead an LLM to adopt a computational path analogous to the human brain's remain unknown.

**Approach.**   To address these issues, we analyze the temporally resolved brain signals of healthy individuals recorded with magnetoencephalography (MEG) [10], while they listened to 10 hours of audiobooks. We then systematically analyze these neural dynamics in conjunction with a benchmark of 17 LLMs varying in architecture, size and training choices.

# 2   Methods

**Problem formalization.** We compare (i) the representations of the human brain in response to natural speech, to (ii) the representations of LLMs in response to the corresponding textual input. Brain activity, here measured with magnetoencephalography (MEG), leads to a high-dimensional time series that depends, in part, on speech input. To test whether the order of computations is similar between the brain and LLMs, we test whether the orders in which representations are generated are correlated between the two systems.

**Linear mapping.** Following others [11, 12, 13], we operationally define a neural "representation" as "linearly readable information". As there is no one-to-one alignment between each MEG sensor and each activation of an LLM, we compare the representations across these two systems through a linear mapping. Specifically, we fit a ridge regression to predict LLM activations ($Y \in \mathbb{R}^{w \times d}$) from brain activity ($X_t \in \mathbb{R}^{w \times s}$):

$$\hat{W} = \arg\min_{W} \left\{ \|Y - XW\|_2^2 + \lambda\|W\|_2^2 \right\}$$

with $w$ the number of words, $d$ the number of LLM activations, $s$ the number of MEG sensors and $t$ the time point relative to word onset. For this, we use scikit-learn's `RidgeCV`, with tuning of logarithmically spaced regularization strength through a grid search approach ($\alpha = 10^{-4}$ to $10^8$, tuned for each dimension independently).

**Alignment score.** To evaluate this alignment score between an LLM and a human brain, we compute, for each time sample relative to word onset, a Pearson correlation between $Y$ and $WX_t$ on a held-out test set. We repeat this procedure across all five train–test folds of the cross-validation.

**Temporal alignment.** After computing this LLM-brain alignment score of $l = 9$ equally spaced layers of the LLM, we evaluate whether the time at which the score peaks correlates with the depth of the layer in the LLM considered. We refer to this as *temporal alignment*. Specifically, we compute, for each layer $T_{\max}$, the mean of the temporal window during which $\tilde{R} \geq 95\%$, where $\tilde{R}$ is the normalized alignment score of the layer, obtained by dividing the alignment score by its maximum value across time. Finally, we compute the Pearson correlation between the $T_{\max}$ and the relative depth of the 9 layers. The result of this correlation is hereafter referred to as *temporal score*.

**Brain data and preprocessing.** We here focus on a large within-subject MEG dataset publicly available [10]. This dataset consists of three healthy participants who listened to 10 h of audio books

in a CTF MEG scanner. To limit the impact of noise we apply a band-pass filter between 0.1 and 20 Hz, down-sample the signal at 30 Hz, time-lock the brain responses to individual words, and epoch the corresponding neural data between -2.5 s and +3 s relative to word onset using MNE-Python [14]. Finally, we z-score MEG signals across words, for each MEG channel and each time point independently.

**LLM activations and preprocessing.** We use a selection of SOTA LLMs to ensure a comprehensive evaluation ranging through architectures, scales, design and training choices. Specifically, we benchmark models such as Llama-3-8B [15, 16], Llama-3.2 (1B, 3B) [15, 17, 18], Mistral-7B-v0.1 [19], Gemma-7B [20], Qwen1.5-7B [21], and GPT-2-XL [22, 23], the latter serving as a historical reference. We also investigate dynamics at play in SOTA state space models: Mamba-1.4B-hf [24, 25] and RecurrentGemma-9B [26]. Additionally, we leverage the Pythia family [27] consisting of 8 models of increasing size and same training setup. Except when explicitly stated, context length is 50 words. For each LLM, we investigate 9 layers linearly distributed between 10% and 90% of the model hierarchy. To mitigate the issue of heterogeneous sizes, we transform these activation patterns with a Principal Component Analysis (n=50) with scikit-learn [28].

**Text preprocessing.** To ensure the processing of the most semantically meaningful words, we study only content words (as opposed to function words), specifically those which belong to the following part-of-speech categories as defined by Spacy [29]: NOUN, VERB, ADJ, ADV. We ensure the replicability of findings for all words (function and content) via control figures in App. F.

**Compute resources.** Evaluating the largest LLM on our dataset to extract both activation patterns for all 9 layers and next-word probability requires 3 V100 GPU-hours and 12 CPU-hours. Pre-processing both LLMs' and brain's activations, fitting and testing a decoding model on our dataset, for each layer and each time step along our temporal window of 5.5s requires in total 4 hours x 9 layers x 15CPUs (Allocated memory: 100 GB) x 3 subjects = 1,620 CPU-hours - for the largest LLM. In total, experiments on 17 LLMs and 9 varying contexts required ~75 GPU-hours and ~42,500 CPU-hours. An internal cluster was used for all experiments. Licenses for the models and dataset used are presented in App. J.

## 3   Results

**Alignment score.**   We compare the representations of LLMs to those of the human brain in response to natural speech. For this, we fit a linear model to predict the LLMs' contextual representations from the MEG activations, time locked to word onset, and we evaluate this linear mapping with a correlation between true and predicted activations on a held-out test set. The results show that the alignment scores between each layer of the LLMs and the brain increase around 0.4s post-word onset (Fig. 1A-C).

**Temporal alignment.**   We next examine the existence of brain-like dynamics of computations in the nine LLMs specified above. On average, these models show a "temporal score" of $r = 0.99$ ($p <$ 1e-06), between the depth of their layers and the MEG responses to words (Fig. 1A-B). This temporal alignment is observed in all studied models, even non-transformer models like ReccurrentGemma-9B and Mamba-1.4B. (Fig. 2C-D). Oldest model GPT2-XL exhibits the lowest Temporal score, though significant, $r = 0.85$. While being grounded in the same seminal transformer design, GPT2-XL is smaller and lacks modern advancements present in the other studied transformers [30, 31]. When not pretrained, the LLMs do not show this alignment, and encode very poorly the brain activity. Together, these results suggest that the order of computations activated through recent LLMs' layers is similar to the order of computations of the human brain listening to natural speech.

**Impact of causality.**   To assess the impact of contextual directionality, we compare two bidirectional LLMs - BERT [32] and RoBERTa [33] - and one bidirectional speech model - Wav2vec2.0 [34] - to the previous causal models, more faithful to the brain's causal mechanisms of language processing. While alignment scores are comparable, the temporal scores of these bidirectional models are substantially lower than those of causal LLMs (see App. B).

**Impact of model size.**   Various architectures, scales and training choices all yield above-chance representational and temporal alignments. To identify the factors that impact the emergence of this phenomenon, we repeat these analyses on a family of LLMs that solely vary in model size. For this,

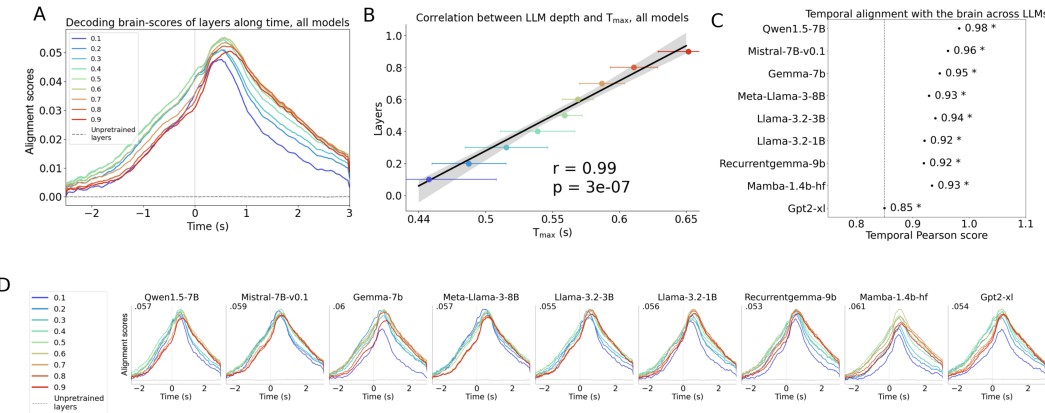

Figure 2: **Human brain and LLMs exhibit temporal alignment.** Correlation between time of peaking alignment scores ($T_{max}$, x-axis) and layer depth shows a highly significant temporal alignment. A. Alignment scores of 9 representative layers across each of the 9 studied LLMs, as a function of word-onset (t=0). Alignment scores have been averaged across models. In dashed gray curves, layers from unpretrained versions of these models, averaged over models. B. The time steps of peaking alignment scores ($T_{max}$, x-axis) are plotted for each representative layer (y-axis), averaged across models. The Temporal score $r$ and associated $p$ are printed on the figure. The grey area indicates the confidence intervals of the regression estimate. Error bars across subjects could not be computed due to the low number of subjects and the need to average across subjects to denoise neural data, though we ensure reproducibility of results across subjects in App. G,C. Here, colored error bars indicate standard deviations of the layer-wise distributions of $T_{max}$ across the 9 presented models. C. Temporal scores are computed and presented for each model studied independently. An asterix next to the Temporal score indicates the score is significant with $p < 5e\text{-}3$. D. Alignment scores of 9 representative layers across each of the 9 presented LLMs, as a function of word-onset (t=0). Each figure presents one model studied independently.

we leverage the Pythia family [27]: eight models of increasing scales. These models are trained in the same way, on the same data, in a highly controlled setup where only model size varies. We find that both the representational and the temporal alignment increase with model size (Fig. 3), from a non-significant temporal score ($r = 0.44$, $p > 0.05$) for the smallest model of 14M parameters, to a highly significant temporal score ($r = 0.96$, $p < 1e\text{-}4$) for the biggest model of 12 billion parameters. The correlation between the temporal score and the log model size reaches $r = 0.87$ ($p = 0.01$). This emergence follows a logarithmic trend, where the temporal and alignment scores of the biggest models tend to plateau. A similar trend is observed for alignment score.

**Impact of context size.** When listening and processing language, humans accumulate narrative context in the form of evidence to extract the richest meaning out of the currently heard word, and anticipate the next one [35]. Motivated by this incremental nature of human language processing, we postulate that brain-like inference dynamics in LLMs emerge with context size. To test how context size impacts representational and temporal alignments, we repeat our analyses on a single model (Llama-3.2 3B) while varying the amount of words in the input context. The results show that representational and temporal alignments increase with context size ($r = 0.81$, $p < 5e\text{-}2$, Fig. 4), from a non-significant temporal score without context ($r = 0.19$, $p > 0.5$) to a highly significant temporal score for context of 1000 words ($r = 0.93$, $p = 3e\text{-}4$). This temporal score increases logarithmically and considerably slows down from context lengths of 50 words. A similar - though lower - correlation is observed for alignment score. We find similar logarithmic increases of both temporal and alignment scores for state-space model Mamba (see in App. E).

**Impact of word predictability.** Autoregressive LLMs are trained to predict incoming words/tokens. In a similar way, predictive coding theory suggests that the brain anticipates upcoming words [36]. Could this similarity be the driving factor of temporal alignment? To test this possibility, we evaluate how temporal alignment varies with word's predictability: if this hypothesis was true,

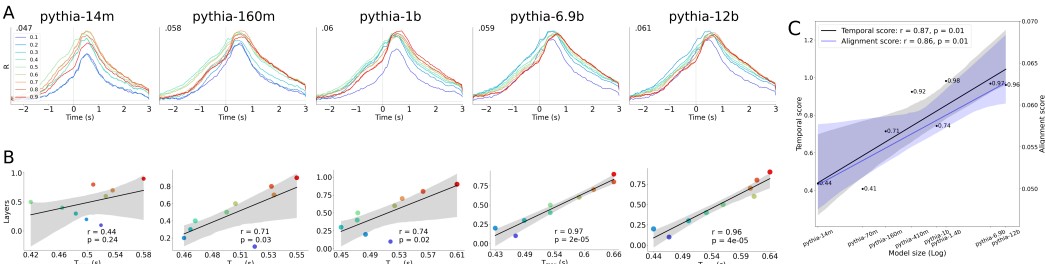

Figure 3: **Temporal alignment emerges with model size.** Colors indicate layer depth. A. Each of the 5 figures on the horizontal axis presents results for a specific model belonging to the Pythia family and studied independently, of size 14m, 160m, 1b, 6.9b and 12b parameters respectively (left to right). The Pythia family hosts 8 models of increasing size, all trained with the same data amount and parameter choices. Figures present evolution of alignment scores $R_{\text{layer}}$ of 9 representative layers, from 10% to 90% of model depth, as a function of word-onset (t=0). B. Each of the 5 figures on the horizontal axis presents results for a specific models belonging to the Pythia family and studied independently. The time steps of peaking alignment scores ($T_{\text{max}}$, x-axis) are plotted for each representative layer (y-axis). The Temporal score $r$ and associated $p$ are printed on the figure. The grey area indicates the confidence intervals of the regression estimate. C. Temporal and alignment scores as functions of model size, for the 8 models forming the Pythia family. The model names (x-axis) are displayed on a logarithmic scale corresponding to their respective size. The Pearson scores $r$ and associated $p$ quantifying these correlations are printed on the figure. The grey and blue areas indicate the confidence intervals of the regression estimates.

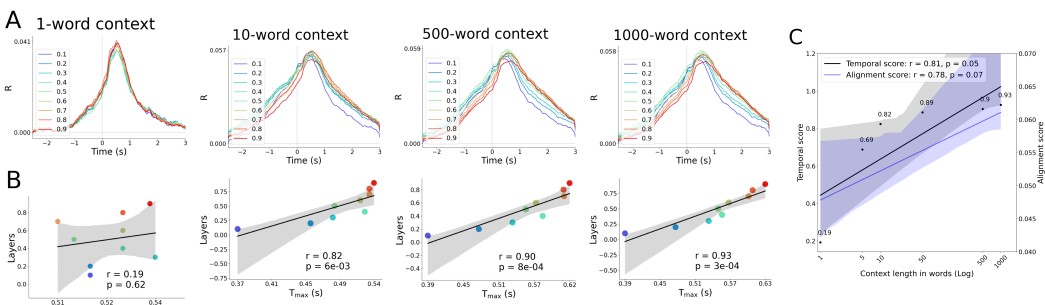

Figure 4: **Temporal alignment increases with the length of the context provided to the LLM.** Colors indicate layer depth. A. Each of the four figures on the horizontal axis presents results for a specific context length provided to Llama-3.2 3B, 1-word, 10-word, 500-word and 1000-word contexts respectively (from left to right). Figures present evolution of alignment scores $R_{\text{layer}}$ of 9 representative layers, from 10% to 90% of model depth, as a function of word-onset (t=0). B. Each of the four figures on the horizontal axis presents results for a specific context length provided to Llama-3.2 3B. The time steps of peaking alignment scores ($T_{\text{max}}$, x-axis) are plotted for each representative layer (y-axis). The Temporal score $r$ and associated $p$ are printed on the figure. The grey area indicates the confidence intervals of the regression estimate. C. Temporal and alignment scores as functions of context length when given to Llama-3.2 3B, for six context lengths (x-axis). Context lengths are displayed on a logarithmic scale. The Pearson scores $r$ and associated $p$ quantifying these correlations are printed on the figure. The grey and blue areas indicate the confidence intervals of the regression estimates.

highly unpredictable words should exhibit low temporal alignment. To control for the fact that predictability can be impacted by the word being a content or function word, for this analysis we include all words of our dataset. For this, we first retrieve the predictability of each word in its context from the softmax-transformed logits of Llama-3-8B. We then separate these words into four

predictability quartiles. Finally, we evaluate temporal alignment for each quartile independently. The results show that most and least predictable words both lead to above chance representational and temporal alignments: $r = 0.92$, $p < 1e\text{-}3$ for the quartile "most expected" and $r = 0.83$, $p < 1e\text{-}2$ for the quartile "most surprising", respectively (Fig. 5). The temporal score for the most surprising quartile does exhibit a lower $r$ value and $p$. However, when computing the Pearson correlation of the layer-wise differences of $T_{\max}$ between "most expected" and "most surprising" quartiles, we do not find a significant impact of contextual predictability on Temporal score ($p = 0.61$, Fig. 5E). This result holds with more layers too (see App. H). Overall, this control analysis indicates that word-predictability alone does not explain the emergence of temporal alignments between LLMs and the brain.

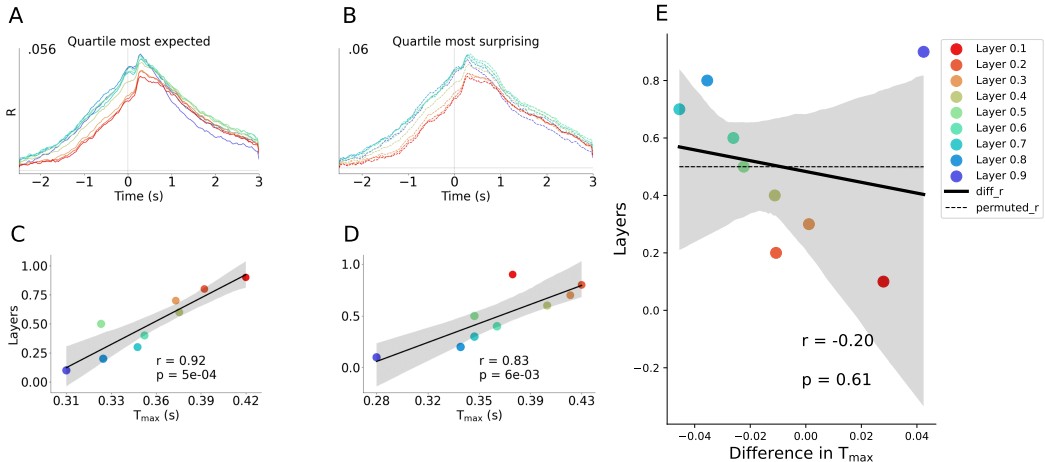

Figure 5: **Temporal alignment holds independently of word predictability.** Colors indicate layer depth. A. Alignment scores of 9 representative layers of Llama-3-8B, from 10% to 90% of layer depth, as a function of word-onset (t=0). Alignment score dynamic curves resulting from evaluating only the quartile of most expected words (from context) among the ~270 000 words forming the dataset. The contextual predictability of words is computed through Llama-3-8B. B. The time steps of peaking alignment scores ($T_{\max}$, x-axis) of the quartile of most expected words are plotted for each representative layer (y-axis) of Llama-3-8B. The Temporal score $r$ and associated $p$ are printed on the figure. The grey area indicates the confidence intervals of the regression estimate. C. Alignment scores of 9 representative layers of Llama-3-8B, from 10% to 90% of layer depth, as a function of word-onset (t=0). Alignment score dynamic curves resulting from evaluating only the quartile of least expected (i.e. more surprising) words from context, among the ~270 000 words forming the dataset. The contextual predictability of words is computed through Llama-3-8B. D. The time steps of peaking alignment scores ($T_{\max}$, x-axis) of the quartile of least expected words are plotted for each representative layer (y-axis) of Llama-3-8B. The Temporal score $r$ and associated $p$ are printed on the figure. The grey area indicates the confidence intervals of the regression estimate. E. The pairwise differences between time steps of peaking alignment scores (Difference in $T_{\max}$, x-axis) per layer, between the quartile of most expected words and the quartile of least expected words, for each representative layer (y-axis) of Llama-3-8B. The Pearson score $r$ and associated $p$ quantifying this correlation are printed on the figure. The grey area indicates the confidence intervals of the regression estimate.

**Correlation between temporal and alignment scores.** We investigate whether, and why, temporal and alignment scores are correlated, while arising from different measurements. We study jointly the temporal scores across all previously studied models - clustered by families - and context lengths, with their respective maximal alignment score – i.e. through their best predictive layer. When plotting both alignment and temporal scores across families of models (Fig. 6A), we find that both alignment and temporal scores increase with model size and context length. When correlating both alignment and temporal scores, we find that temporal score indeed correlates significantly with alignment score (Pearson score = 0.54, $p = 9e\text{-}04$) (Fig. 6B). This result indicates that the capacity of an

LLM to predict neural signals of the human brain is correlated to how tightly aligned its pathway of computations is with the one at play in the brain, when processing language.

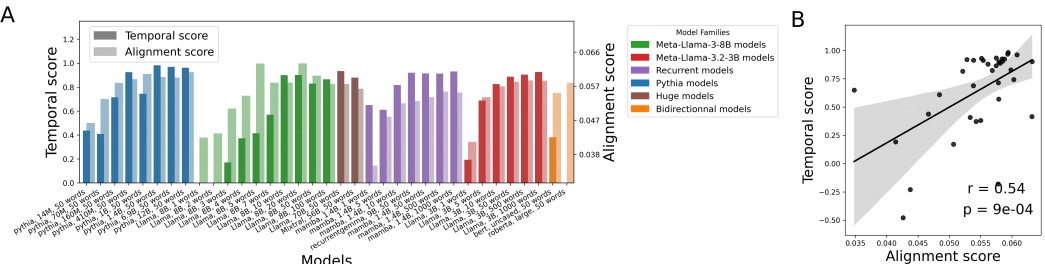

Figure 6: **Temporal and alignment scores are significantly correlated.** A. Temporal score and alignment score evolve similarly across models and context lengths, along the x-axis, clustered by families of models. Models are presented in the format [model name, model size, context length]. B. Temporal score (y-axis) and alignment score (x-axis) are significantly correlated over all models and variance in contexts size presented in this research. The Pearson score $r$ and associated $p$ quantifying this correlation are printed on the figure. The grey area indicates the confidence interval of the regression estimate.

## 4 Discussion

**Temporal alignment.** This work investigates whether and when LLMs generate representations in an order similar to the human brain's during natural speech listening. This study provides three main contributions. First, the order of representations generated by LLMs' layers strongly correlates with the sequence of neural activations observed in the human brain recorded with MEG. This *temporal* alignment complements previous work on the *anatomical* alignment observed between LLMs' layers and functional regions of the human brain [5, 8, 3, 9, 1], and systematize early report of temporal alignment between language models and the brain [4, 6].

**A shared computational path.** We show that temporal scores is independent of the word predictability (Fig. 5). Beyond indicating the autoregressive generation capacities of the LLM, the temporal score thus seems to indicate its brain-like inference dynamics. Additionally, as previously reported [2, 4, 3, 37], the best alignment scores are achieved in the intermediary layers. This result suggests that the intermediary representations (layer=0.6 of architecture) – as opposed to the input and the prediction – are best aligned between brains and LLMs. Finally, while part of the alignment score can be explained by the latent representations present from the very first layer — accounting for the relatively high alignment score observed from the first layers of the LLM — the temporal score accounts for evolving representations, a sequentially ordered mechanism of processing. Together, these results indicate that the brain and LLMs don't just share similar representations, but a similar computational path too. The increase of alignment score in more brain-like models may stem from a shared sequential structure of computations between the LLMs and the brain, as captured by the temporal score.

**Impact of model architecture.** Second, this LLM-brain alignment is observed in non-transformer architectures, such as state-space models. While other architectures (e.g. fully connected networks [38], Kolmogorov-Arnold Networks [39]) remain to be investigated, this finding suggests that the convergence between LLMs and the brain is architecture-independent. If confirmed, this result provides additional evidence that alignment score does not result from a trivial inductive bias. It remains to be investigated, however, whether it relates to model objective (next token prediction) or to the structure of language [40].

**Impact of context and model size.** Third, our experiments show that this alignment directly depends on (i) context size and (ii) model size – although with a saturation beyond 70B parameters models (App. D). These MEG results extend previous works on scaling laws in neuro-ai [37, 41, 42].

In particular, [37, 43] showed that LLMs that best predict functional magnetic resonance imaging (fMRI) responses to natural speech are those with the largest amount of parameters. In parallel, [44, 45] showed that context size improved the alignment between brain and fMRI. Here, we further show the effect of context size and model size to increase logarithmically, hence pointing to diminishing returns, if not a plateau. Extending our analyses to larger LLMs, e.g. Llama 3.3 70B, does not yield major improvement as compared to smaller LLMs e.g. Llama 3.2 3B, hence pointing to a plateau effect of scaling laws identifiable with MEG. Together, these findings clarify the specific conditions required for brain-like representations and computations to emerge. It remains unclear, however, whether context and size act directly on the alignment, or are confounded by other uncontrolled variables, such as linguistic performance. For example, in App. I, we find that performance is correlated with temporal alignment for specific conditions - LLM with increasing context lengths - but not others - LLMs with increasing sizes. Disentangling the causal chain that links these factors remains a major research avenue.

**Working memory.** Interestingly, the comparison between State Space Models (SSMs) and LLMs offers a new perspective to investigate context-size. Indeed, transformers compute contextual representations, thanks to a non linear combination of the past context, and hence require very large memory buffers that are implausible in the brain. By contrast, SSMs can be thought of as recurrent neural networks (RNNs) with hidden states that linearly evolve over time. At each time point, they thus represent the full context with their hidden state – an approach presumably similar to the brain. Our results show that SSMs do exhibit modest yet consistent improvements in temporal scores even at very large context sizes (App. E), and thus show that it is, in fact possible, to build and maintain a long context in a single hidden state. However, further research remains necessary to evaluate the degradation of such memory in the absence of meaningful context, such as during the memorization of a random digit sequence like a phone number – a task recognized as highly constrained in human subjects [46].

**Limitations.** Three main limitations should be highlighted. First, MEG has a limited spatial resolution of brain activity. This recording device does not allow a single-neuron recording, and is largely unable to pick up deep sources. Consequently, the precise neuronal bases of the present findings remain to be further explored with intracranial recordings. Second, the present dataset only consists of 3 subjects. This design choice was motivated by the fact that subjects each listen to 10 h of audiobooks, making this dataset the largest per-subject MEG listening dataset authors found, among multi-subject datasets. As alignment models are trained subject-wise, the biggest possible per-subject amount of data allows the most precise brain-alignment with LLMs dynamics. While App. G shows similar findings can be identified within each of these three subjects, how these alignments *vary across individuals* thus remains unknown. Finally, the present work only focuses on pretrained text-models. Yet, humans necessarily process language through sensory modality. Consequently, future work remains necessary to investigate the similarities between brains and speech models [47, 48, 5, 9].

**Impact.** The present findings reveal striking similarities between human brain responses and sequential representations in multiple kinds of large language models (LLMs). Yet, their architecture, sensory modality, learning objective, and training regime are remarkably disjoint [49, 50, 51, 52]. Furthermore, specific features like syntactic structures and semantic roles may be represented fundamentally differently [53, 54]. That partially converging computational paths arise despite these differences is thus all the more surprising, and highlights the necessity to clarify the computational principles that lead language processing to be partially shared between biological and artificial systems. We hope that systematically investigating the factors that steer LLMs to function similarly to the brain will help reduce this major gap, and ultimately chart a path toward building artificial systems that learn as efficiently as the human brain.

## 5 Acknowledgments

We warmly thank the authors of [10] for building the MEG dataset on which we relied for this study.

We wish to thank every member of Meta's legal team for helping us to process quickly through the legal process in order to submit in time to the Neurips 2025 conference.

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

# Supplementary Material and Technical Appendices

## A Encoding scores for two LLMs.

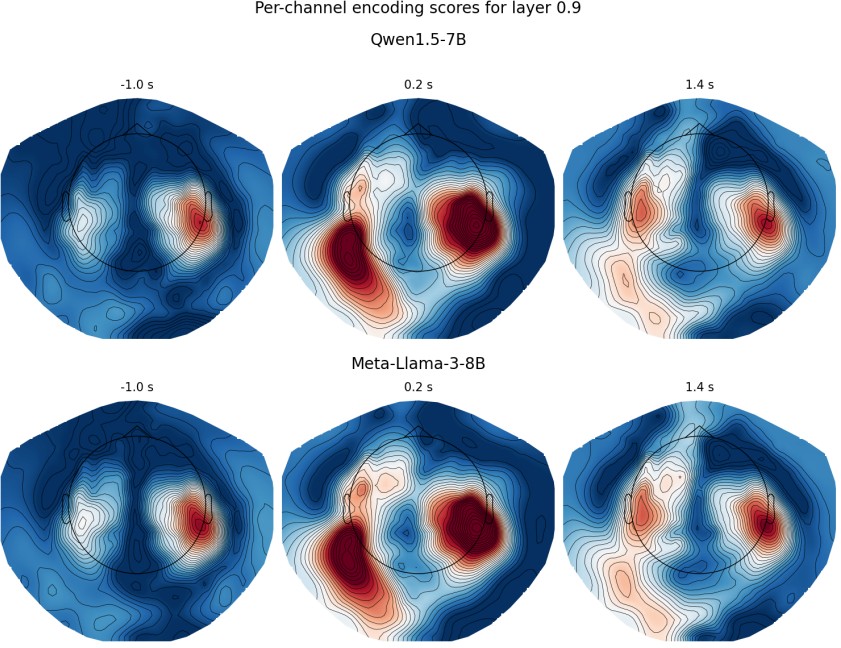

Figure 7: **Encoding scores for two LLMs.** Per-channel encoding scores for the layer l=0.9 of two LLMs, Qwen 1.5-7B (up) and LLama-3-8B (down), at several time points during word prediction pre word-onset (-1.0s, left), and processing post word onset (0.2s middle, and 1.4s right).

# B   Temporal and alignment scores of bidirectional models.

Below are presented results for bidirectional LLMs BERT and RoBERTa, alongside bidirectional speech model Wav2vec2. All three models show alignment scores comparable to larger, more recent and causal LLMs, but much lower, non-significant temporal scores.

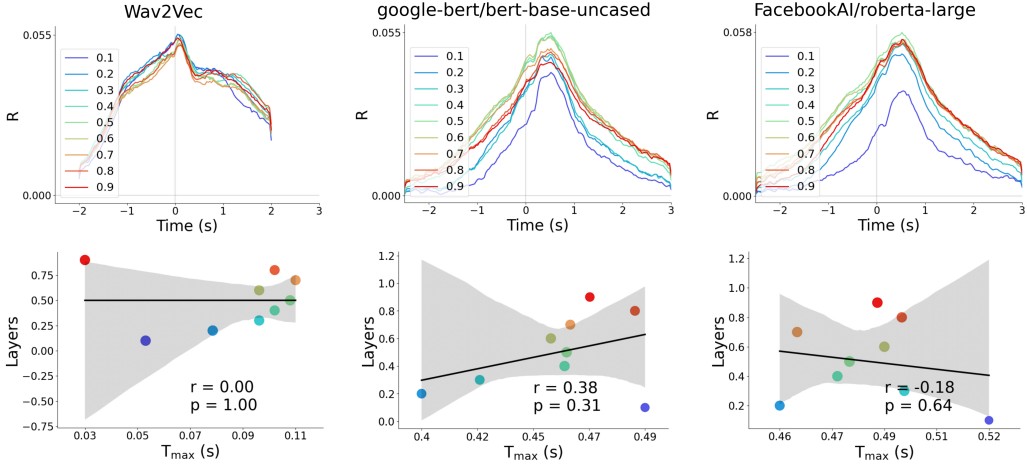

Figure 8: **Temporal and alignment scores of bidirectional models.** Each of the 3 figures presents results for a Wav2vec2, BERT and RoBERTa. Upper figures on the horizontal axis present evolution of alignment scores $R_{\text{layer}}$ of 9 representative layers, from 10% to 90% of model depth, as a function of word-onset (t=0). Lower figures on the horizontal axis presents the time steps of peaking alignment scores ($T_{\text{max}}$, x-axis) plotted against each representative layer (y-axis). The Temporal score $r$ and associated $p$ are printed on the figure. The grey area indicates the confidence intervals of the regression estimate.

# C  Alignment scores across subjects, per LLM

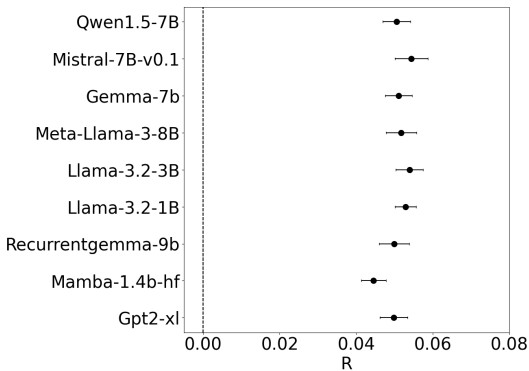

Figure 9: **Alignment scores across subjects, per LLM.** Maximum alignment scores per model, Each errorbar is a standard deviation across the three subjects of the dataset.

# D   Temporal and alignment scores of the largest models

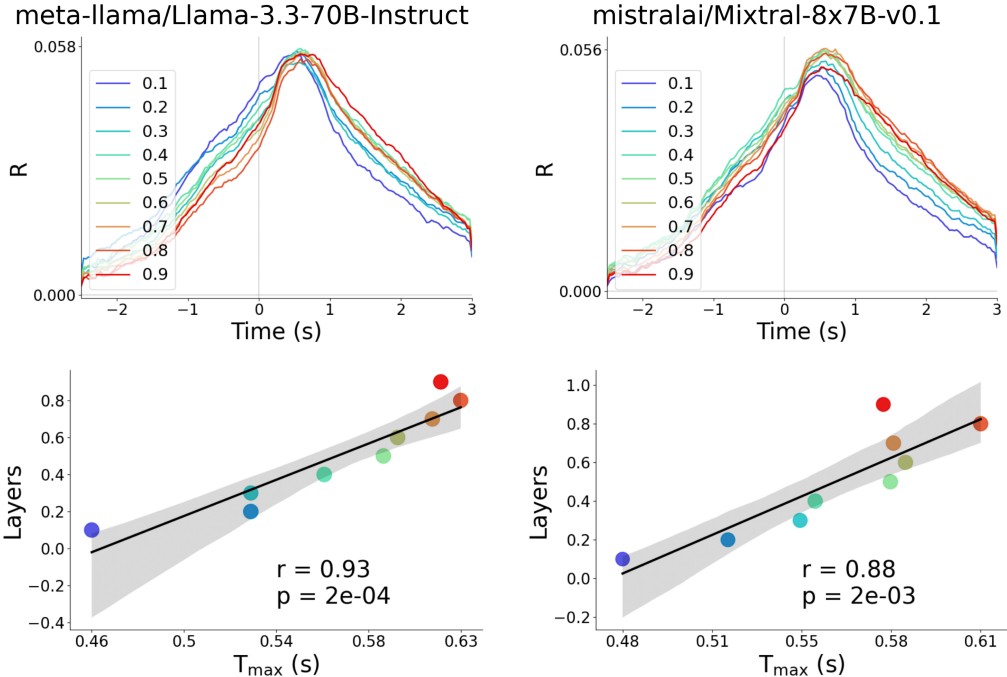

Figure 10: **Temporal and alignment scores of the largest models.** Each of the 2 figures presents results for a Llama3.3-70B-Instruct and Mixtral-8x7B-v0.1. Upper figures on the horizontal axis present evolution of alignment scores $R_{\text{layer}}$ of 9 representative layers, from 10% to 90% of model depth, as a function of word-onset (t=0). Lower figures on the horizontal axis presents the time steps of peaking alignment scores ($T_{\text{max}}$, x-axis) plotted against each representative layer (y-axis). The Temporal score $r$ and associated $p$ are printed on the figure. The grey area indicates the confidence intervals of the regression estimate.

# E  Temporal alignment increases with the length of the context provided to State-Space Model Mamba-1.4B

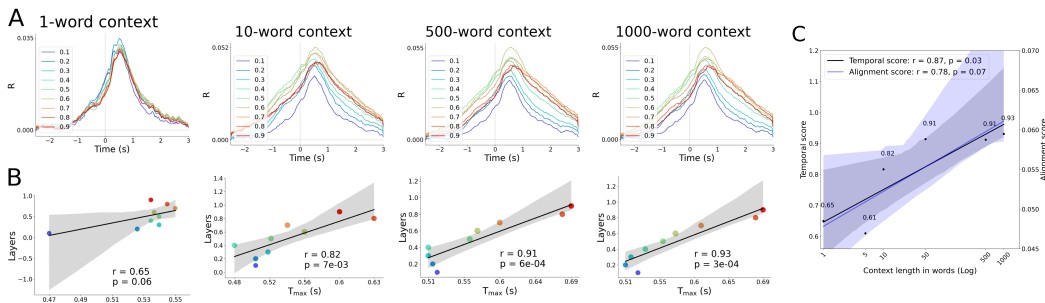

Figure 11: **Temporal alignment increases with the length of the context provided to State-Space Model Mamba-1.4B.** Colors indicate layer depth. A. Each of the four figures on the horizontal axis presents results for a specific context length provided to State-Space Model Mamba-1.4B, 1-word, 10-word, 500-word and 1000-word contexts respectively (from left to right). Figures present evolution of alignment scores $R_{\text{layer}}$ of 9 representative layers, from 10% to 90% of model depth, as a function of word-onset (t=0). B. Each of the four figures on the horizontal axis presents results for a specific context length provided to Mamba-1.4B. The time steps of peaking alignment scores ($T_{\text{max}}$, x-axis) are plotted for each representative layer (y-axis). The Temporal score $r$ and associated $p$ are printed on the figure. The grey area indicates the confidence intervals of the regression estimate. C. Temporal and alignment scores as functions of context length when given to Mamba-1.4B, for six context lengths (x-axis). Context lengths are displayed on a logarithmic scale. The Pearson scores $r$ and associated $p$ quantifying these correlations are printed on the figure. The grey and blue areas indicate the confidence intervals of the regression estimates.

## F Human brain and LLMs exhibit temporal alignment. Analysis performed for all types of words: function and content words

To ensure the processing of the most semantically meaningful words, we often study only content words in the main text of this paper (as opposed to function words), specifically those which belong to the following part-of-speech categories as defined by Spacy: NOUN, VERB, ADJ, ADV. Here, we test for replicability of findings for all words (function and content).

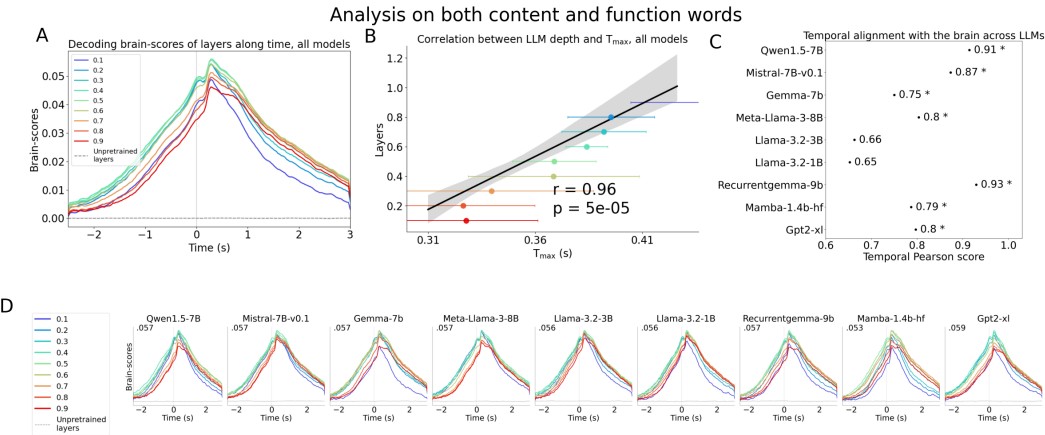

Figure 12: **Human brain and LLMs exhibit temporal alignment. Analysis performed for all types of words: function and content words.** Correlation between time of peaking alignment scores ($T_{max}$, x-axis) and layer depth shows a highly significant temporal alignment. A. Alignment scores of 9 representative layers across each of the 9 studied LLMs, as a function of word-onset (t=0). Alignment scores have been averaged across models. B. The time steps of peaking alignment scores ($T_{max}$, x-axis) are plotted for each representative layer (y-axis), averaged across models. The Temporal score $r$ and associated $P_{value}$ are printed on the figure. The grey area indicates the confidence intervals of the regression estimate. Here, colored error bars indicate standard deviations of the layer-wise distributions of $T_{max}$ across the 9 presented models. C. Temporal scores are computed and presented for each model studied independently. An asterix next to the Temporal score indicates the score is significant with $P_{value} < 5e$-3. D. Alignment scores of 9 representative layers across each of the 9 presented LLMs, as a function of word-onset (t=0). Each figure presents one model studied independently.

# G   Human brain and LLMs exhibit temporal alignment – Subject 1, 2 and 3 plotted individually

We ensure reproducibility of results presented in the main text of this paper – averaged across subjects – this time for individual subjects.

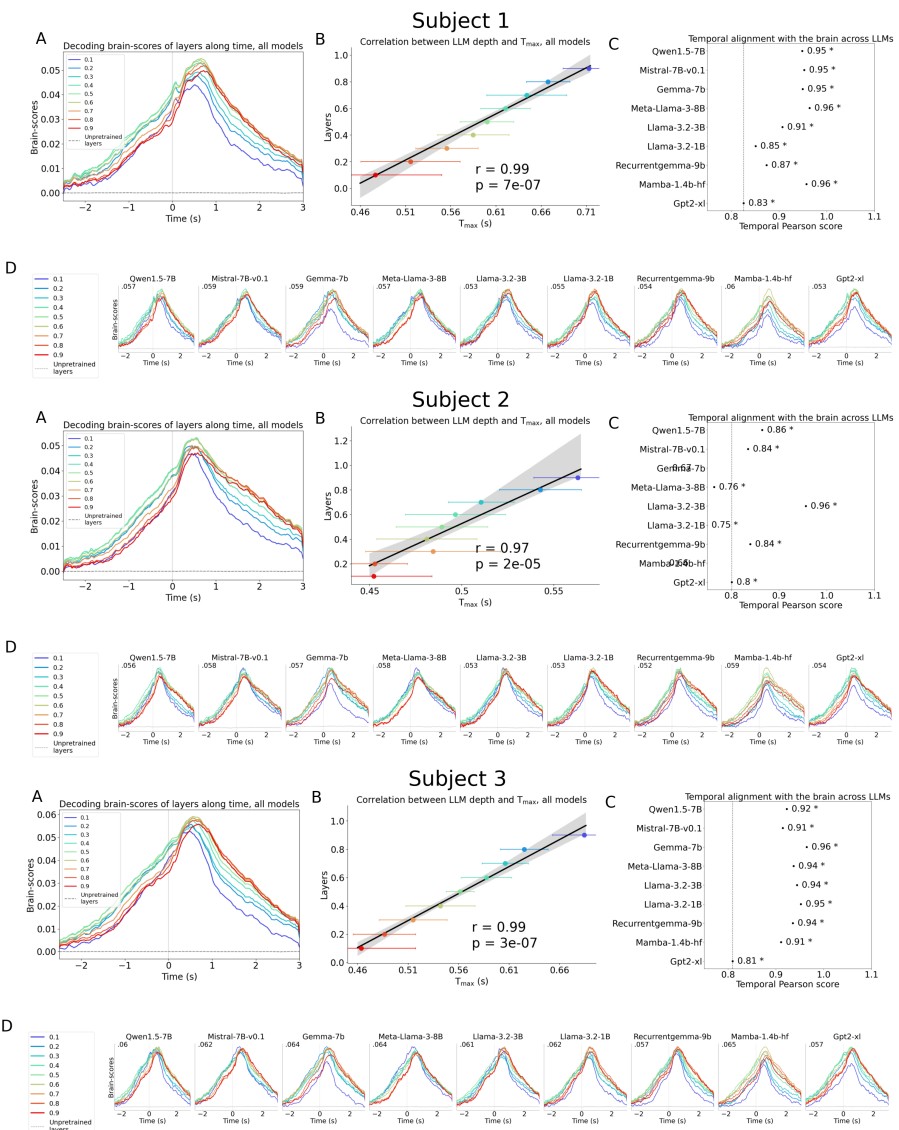

Figure 13: **Human brain and LLMs exhibit temporal alignment – Subject 1, 2 and 3 plotted individually**
Correlation between time of peaking alignment scores ($T_{max}$, x-axis) and layer depth shows a highly significant temporal alignment. A. Alignment scores of 9 representative layers across each of the 9 studied LLMs, as a function of word-onset (t=0). Alignment scores have been averaged across models. B. The time steps of peaking alignment scores ($T_{max}$, x-axis) are plotted for each representative layer (y-axis), averaged across models. The Temporal score $r$ and associated $P_{value}$ are printed on the figure. The grey area indicates the confidence intervals of the regression estimate. Here, colored error bars indicate standard deviations of the layer-wise distributions of $T_{max}$ across the 9 presented models. C. Temporal scores are computed and presented for each model studied independently. An asterix next to the Temporal score indicates the score is significant with $P_{value} < 5e\text{-}3$. D. Alignment scores of 9 representative layers across each of the 9 presented LLMs, as a function of word-onset (t=0). Each figure presents one model studied independently.

## H Temporal alignment holds independently of word predictability

When computing the Pearson correlation of the layer-wise differences of $T_{\max}$ between "most expected" and "most surprising" quartiles, we do not find a significant impact of contextual predictability on Temporal score. To ensure this result, we perform here the same analysis on twice as many layers across the depth of Llama-3-8B architecture.

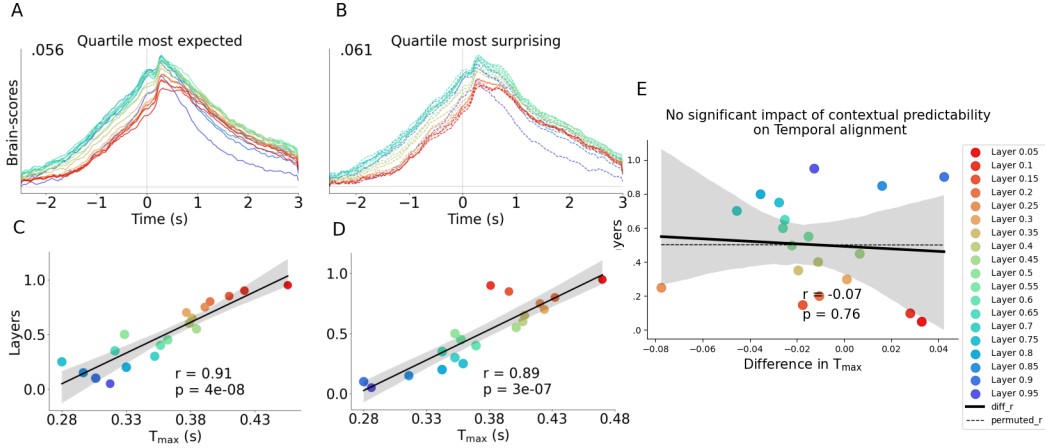

Figure 14: **Temporal alignment holds independently of word predictability.** Colors indicate layer depth. A. Alignment scores of 19 representative layers of Llama-3-8B, from 10% to 90% of layer depth, as a function of word-onset (t=0). Alignment score dynamic curves resulting from evaluating only the quartile of most expected words (from context) among the ~270 000 words forming the dataset. The contextual predictability of words is computed through Llama-3-8B. B. The time steps of peaking alignment scores ($T_{\max}$, x-axis) of the quartile of most expected words are plotted for each representative layer (y-axis) of Llama-3-8B. The Temporal score $r$ and associated $P_{\mathrm{value}}$ are printed on the figure. The grey area indicates the confidence intervals of the regression estimate. C. Alignment scores of 19 representative layers of Llama-3-8B, from 10% to 90% of layer depth, as a function of word-onset (t=0). Alignment score dynamic curves resulting from evaluating only the quartile of least expected (i.e. more surprising) words from context, among the ~270 000 words forming the dataset. The contextual predictability of words is computed through Llama-3-8B. D. The time steps of peaking alignment scores ($T_{\max}$, x-axis) of the quartile of least expected words are plotted for each representative layer (y-axis) of Llama-3-8B. The Temporal score $r$ and associated $P_{\mathrm{value}}$ are printed on the figure. The grey area indicates the confidence intervals of the regression estimate. E. The pairwise differences between time steps of peaking alignment scores (Difference in $T_{\max}$, x-axis) per layer, between the quartile of most expected words and the quartile of least expected words, for each representative layer (y-axis) of Llama-3-8B. The Pearson score $r$ and associated $P_{\mathrm{value}}$ quantifying this correlation are printed on the figure. The grey area indicates the confidence intervals of the regression estimate.

# I  Temporal scores along top-3 accuracies evaluated on our dataset for next-token prediction across LLMs

We find that performance is correlated with temporal alignment for specific conditions – LLMs with increasing context lengths – but not others – LLMs with increasing sizes. Regression estimates illustrate how scaling model size or context affect temporal alignment with brain activity. Increasing context lengths significantly correlates with temporal alignment (p = 0.01). Increasing model sizes does not significantly correlate with temporal alignment.

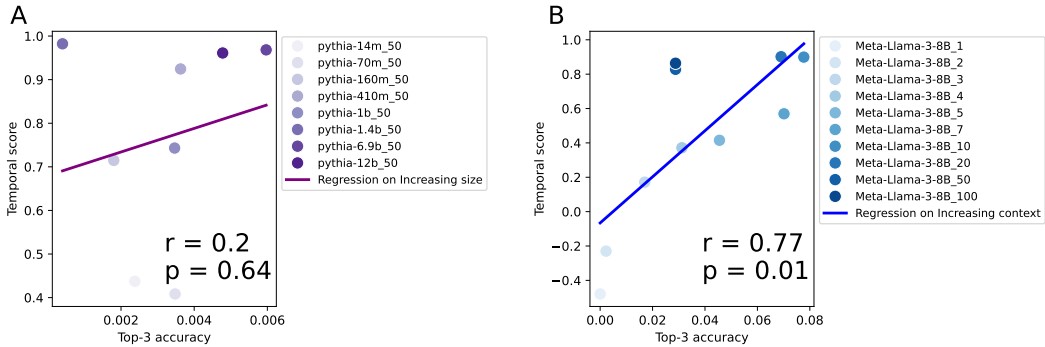

Figure 15: **Temporal scores along top-3 accuracies evaluated on our dataset for next-token prediction across LLMs**. A and B show the correlation between Temporal score (y-axis) and top-k accuracy (x-axis), across two subsets of models: A. Pythia models of increasing size, evaluated at top-k = 3. B. Meta-Llama-3-8B model with increasing context lengths, evaluated at top-k = 3.

## J Licenses

The dataset used in this paper is licensed under a Creative Commons Attribution 4.0 International License.
Regarding the models used, here follows a list of their licenses:

- Qwen/Qwen1.5-7B: Tongyi Qianwen License
- mistralai/Mistral-7B-v0.1: Apache 2.0 License
- google/gemma-7b: Gemma License
- meta-llama/Meta-Llama-3-8B: Llama 3 Community License Agreement
- meta-llama/Llama-3.2-3B: Llama 3.2 Community License Agreement
- meta-llama/Llama-3.2-1B: Llama 3.2 Community License Agreement
- google/recurrentgemma-9b: Gemma License
- state-spaces/mamba-1.4b-hf: Apache 2.0 License
- openai-community/gpt2-xl: MIT License
- Pythia family of models: Apache 2.0 License

