# OpenReview forum: "Scaling and context steer LLMs along the same computational path as the human brain"
_NeurIPS.cc/2025/Conference — NeurIPS 2025 spotlight_

### Official Review · Reviewer_eQDJ · 2025-06-09

**Clarity:** 3
**Significance:** 3
**Originality:** 2
**Rating:** 5
**Confidence:** 3

**Summary:**

The paper focuses on the temporal alignment of LLMs to human brain MEG responses, i.e., later model layers show greatest representational similarity to neural responses at greater times past word onset compared to earlier model layers (temporal alignment). They conduct many analyses to study how temporal alignment of LLMs is related to overall brain alignment, context size, and model size.

**Questions:**

None

**Ethical Concerns:**

["NO or VERY MINOR ethics concerns only"]

**Final Justification:**

Weakness 1 -- partially addressed, I think the Pearson scores are still low, lower than the numbers from prior works they cited. Though, it is helpful that they contextualized that MEG Pearson scores are low in general.

Weakness 2 -- mostly addressed by their comment.

**Limitations:**

Yes

**Quality:**

3

**Strengths And Weaknesses:**

Strengths:
- Interesting finding that LLMs exhibit temporal alignment to human brains.
- Interesting findings that temporal alignment is correlated to overall brain alignment, context size, and model size

Weaknesses:
- Pearson correlation of roughly 0.06 as the maximum across different models is quite low, it would be helpful if the paper clarified what the noise ceiling is for the data, to make it easier to understand how well these models are predicting the neural responses.
- "To ensure that the strong Temporal scores we find are not led by this parallel between predicting of next word by the model and hearing of next-word by the human, we study the impact of next words’ predictability on temporal alignment." However, I think one alternative hypothesis is that the model's next-word prediction aligns with the humans' next-word prediction, not next-word hearing. Since deeper model layers build up representations for next-word prediction, it might align more to next-word prediction that humans are doing, i.e., it might simply suggest a similar objective of next-word prediction, not a similar order of computations between LLMs and humans. Having a similar objective would also explain that there is no significant layer-wise difference of T-max between "most expected" and "least expected" quartiles, because humans might also find the model's "most expected" words to be more expected and the model's "least expected" words to be less expected.

---

> ### Author Rebuttal · Authors · 2025-07-30
>
> We thank Reviewer eQDJ for their thorough review. We propose to amend our manuscript to address these issues and add five additional sets of results.
>
> **1. Pearson correlation**
>
>  These Pearson R scores are indeed low, but remain in the range typically observed with electrophysiology. For example, Caucheteux and King (2022) report, from MEG results, brain-scores around R=0.07, peaking at R=0.13 for the best channels. Goldstein et al (2023) report, from intracranial data R=0.2 on the electrodes specifically selected to show language responses. These values are low because (1) the signal-to-noise ratio in electrophysiology is low and (2) there is no denoising procedure (e.g. averaging across repetitions, normalizing by estimated noise ceiling etc). We will add a figure with confidence intervals to illustrate that in spite of being small, these effect sizes are highly reliable.
> - Caucheteux & King (2022), Communications Biology. Brains and algorithms partially converge in natural language processing.
> - Goldstein et al. (2022), BioRxiv. Correspondence between the layered structure of deep language models and temporal structure of natural language processing in the human brain
>
>
> **2. Shared objective vs representations**
>
>  This is an interesting remark. To address it, we now amended the discussion as follows:
>
> As previously reported (Jain et al, (2020), Caucheteux and King (2022), Toneva and Wehbe (2019), Antonello et al. (2023)), the best brain scores are achieved in the intermediary layers. This result suggests that the intermediary representations (layer=60% of architecture) – as opposed to the input and the prediction – are best aligned between brains and LLMs, suggesting that these two systems don’t just share the same objective, but the same computational path too.
>
> - Jain et al. (2020), NeurIPS. "Interpretable multi-timescale models for predicting fMRI responses to continuous natural speech."
> - Caucheteux & King (2022), Communications Biology. "Brains and algorithms partially converge in natural language processing."
> - Toneva & Wehbe (2019), NeurIPS. "Interpreting and improving natural-language processing (in machines) with natural language-processing (in the brain)."
> - Antonello et al 2023  NeurIPS. "Scaling laws for language encoding models in fMRI."
>
>
> **3. New experiments: Additional models and analyses**
>
>
> To address reviewers’ comments, we now run five additional analyses:
>
> 1. To assess the impact of contextual directionality, we compare two bidirectional LLMs (BERT and RoBERTa) to causal models previously showcased in the manuscript, which reflect more closely the brain’s causal processing of language. While brain scores are comparable, the temporal scores of bidirectional models are substantially lower than those of causal LLMs.
>
> 2. We also analyse a speech model, Wav2vec2, bidirectional as well, and compare this model to the other models presented. While brain scores are comparable, the temporal scores of bidirectional Wav2vec2.0 is also substantially lower than those of causal LLMs.
>
> 3. We extend our analyses to a “large LLM”, i.e. Llama 3.3 70B. The results suggest that there is no major improvement as compared to smaller LLMs e.g. Llama 3.2 3B, hence pointing to a plateau effect of scaling laws identifiable with MEG.
>
> 4. We now added GPT2-S (137M) compared to GPT-XL (1.6B) to study scaling in this older family of models.
>
> | Model          | Brain Score | Temporal Score |
> |----------------|-------------|----------------|
> | Wav2Vec2.0     | .055        | .00            |
> | BERT           | .055        | .38            |
> | RoBERTa        | .058        | -.18           |
> | GPT2           | .054        | .20            |
> | GPT2-XL        | .054        | .87            |
> | Mamba          | .061        | .93            |
> | Mistral-7B     | .059        | .96            |
> | LLaMA-3.2-3B   | .055        | .94            |
> | LLaMA-3.3-70B  | .058        | .93            |
>
>
> 5. Finally, we now added analyses on even longer context sizes, with up to 1000 words on both Llama-3.2 and Mamba (State-Space Model).
> The results indicate that up to 1000 words, longer contexts still lead to an increase of temporal alignment, even when the brain-score plateaus.
>
> ### Temporal Score
> | Context length | 1    | 5    | 10   | 50   | 500  | 1000 |
> |----------------|------|------|------|------|------|------|
> | Mamba-1.4b     | .65  | .61  | .82  | .91  | .91  | .93  |
> | LLaMA-3.2-3B   | .19  | .69  | .82  | .89  | .90  | .93  |
>
> ### Brain score
> | Context length | 1     | 5     | 10    | 50    | 500   | 1000 |
> |----------------|-------|-------|-------|-------|-------|-------|
> | Mamba-1.4b     | .035  | .048  | .052  | .054  | .055  | .055  |
> | LLaMA-3.2-3B   | .041  | .054  | .057  | .058  | .059  | .058  |
>
>
>
>
> We would like to thank again Reviewer eQDJ for their insightful comments, and hope this further strengthens our contribution.

---

> > ### Comment · Reviewer_eQDJ · 2025-08-05
> >
> > Weakness 1 -- partially addressed, I think the Pearson scores are still low, lower than the numbers from prior works they cited. Though, it is helpful that they contextualized that MEG Pearson scores are low in general.
> >
> > Weakness 2 -- mostly addressed by their comment.
> >
> > One final thought is a suggestion that the paper should more explicitly highlight what might be the most interesting results, e.g., human brain and LLMs exhibit temporal alignment, temporal alignment emerges with model size, and temporal alignment increases with the length of the context provided to the LLM.  I think the abstract and end of introduction would benefit from explicitly mentioning these results.
> >
> > They could also dedicate more space to highlight how they differ from prior works that showed some initial results on temporal alignment.
> >
> > I have increased my score from 4 to 5.

---

> > > ### Author Response · Authors · 2025-08-06
> > >
> > > Thank you for your comment. We do agree that the abstract and introduction was insufficiently clear on these specific results. Since the submission, we amended our paper to state more explicitly our contributions.
> > > Thank you again for your thorough review.

---

> ### Author Response · Authors · 2025-08-05
>
> Dear reviewer, in light of the added experiments and revised sections, is there any additional element you believe we could add or discuss to address your review?

---

### Official Review · Reviewer_tqrA · 2025-06-30

**Clarity:** 4
**Significance:** 2
**Originality:** 2
**Rating:** 4
**Confidence:** 5

**Summary:**

The paper generates a number of language encoding models for MEG and then measures what they refer to as "temporal alignment" between the models and the brain. In particular they show that earlier layers of language models have earlier alignment peaks (the time at which the encoding performance is highest) than later layers.  They further show that this correspondence is strongest in largest models, scales with context size, and that this temporal alignment score is itself correlated which general encoding performance.

**Questions:**

Please clarify the method of computing brain-score. The existing paragraph is somewhat inadequate.

Since everything appears to be averaged across sensors, it is unclear to me what the spatial patterns driving this temporal alignment are. It would be nicer to see how this temporal alignment shifts on a per-sensor basis. I think this would be more along the lines of something that I might be interested to see. The general lack of spatial specificity in the results here is something that could be improved.

Are the temporal alignment trends observed in the paper a property of LLMs specifically? Do the same results apply to speech models like Whisper or Wav2Vec?

**Ethical Concerns:**

["NO or VERY MINOR ethics concerns only"]

**Final Justification:**

I am satisfied with the author's responses to my concerns. I am especially appreciative of the extended literature discussion involving whether these results should be expected based on prior literature, as well as the promise to include per-sensor results and added scaling results. Contingent on these changes being added to the paper, I am willing to change my score to a 4.

**Limitations:**

See above.

**Quality:**

3

**Strengths And Weaknesses:**

My primary concern here is about the providence of the temporal alignment score that the authors heavily base their paper around. The observation that later layers of LMs model later times in the brain is not really that surprising - the layers are more lexical (and therefore contain information that is earlier in the language processing pipeline) in the early stages of a model and more contextual in the later layers of a model, which is a natural consequence of the hierarchical manner in which LLM architectures build up their representations. In a similar way, auditory cortex (earlier) is more lexical than prefrontal cortex (later).  In addition to being unsurprising, the observation is also not particular novel; a number of the papers that the authors cite contain the same basic trend. The observation that larger models are more "temporally aligned" than smaller models is not something I have explicitly seen mentioned before, but it is also not something I find especially notable, mostly because I would expect larger models to have a more finely-tuned and elongated processing pipeline simply by virtue of being less constrained in parameter count. There is also limited-to-no attention given in the paper to the now fairly extensive literature that already exists regarding scaling laws for encoding models, although I consider this a minor point.

As another minor point, I was somewhat confused about what the brain-score number that is being plotted is. I understand it to be some sort of Pearson correlation metric, however it is unclear how this metric is averaged across sensors. My assumption is that the Pearson correlation is independently computed for each sensor and then averaged across the brain, since this is the general convention in this field, although please correct me if I am wrong.

I do think the paper is very well written and exceptionally clear. I have little to say about obvious scientific mistakes that are made by the authors. Overall, my general opinion is that this paper has no major scientific issues, but that the results are somewhat low impact given that they appear to be something of a replication. I have no problem with replications, but I am also somewhat unsure whether they belong at NeurIPS. I have given the paper a 3.

---

> ### Author Rebuttal · Authors · 2025-07-30
>
> We thank Reviewer tqrA for their thorough review. We propose to amend several sections of the manuscript to address these issues and add five additional sets of results.
>
> **1. Contribution**
>
> We agree that several studies had shown a temporal alignment between brain and LLM before – although it is here demonstrated with 20x more data – 10h per participant, compared to 30 min per subject – and with minimal preprocessing – e.g. no localizer or analyses restricted to clusters or regions of interest – compared to both Li et al. (2023) and Goldstein et al. (2022).
>
> - Li et al. (2023), Nature Neuroscience. Dissecting neural computations in the human auditory pathway using deep neural networks for speech
> - Goldstein et al. (2022), BioRxiv. Correspondence between the layered structure of deep language models and temporal structure of natural language processing in the human brain
>
> Rather, we start from this independent replication to :
>
> **(1)** Systematically compare 23 diversified language models to brain activity.
>
> **(2)** Use this benchmark to identify the *factors* that cause this temporal alignment (architecture, model size, training quantity, directionality, context length).
>
> **(3)** Successfully test the correlation between higher brain scores and higher temporal alignment, identifying conditions where brain score plateaus while temporal score continues to rise (cf "6. New Experiments" for further details).
>
> While we believe this contribution is accurately stated in the introduction and discussion, we do agree that the abstract was insufficiently clear on our exact contribution. We thus propose to amend it as follows:
>
> - “However, *whether* this representational alignment arises from a similar sequence of computations remains elusive. “ -> “However, *whether and why* this representational alignment arises...”
> - “Our analyses *reveal* that LLMs and the brain generate representations in a similar order” -> “Our analyses *confirm* ...”
> - “Overall, the alignment between LLMs and the brain provides novel elements supporting a partial convergence between biological and artificial neural networks.” -> “Overall, this study *clarifies the factors underlying the partial convergence* between ...”
>
> **2. Credit to scaling laws studies**
>
>  We agree that we insufficiently credited previous work on scaling laws. We now amended the discussion as follows:
>
> Third, our experiments show that this alignment directly depends on (i) context size and (ii) model size. These MEG results extend previous works on scaling laws in neuro-ai (Antonello et al. (2023), Bonnasse-Gahot & Pallier (2024), Tikochinski et al. (2025), d’Ascoli et al. (2024), Banville et al. (2025), Caucheteux et al. (2023)).
> In particular, Antonello et al. (2023) and Bonnasse-Gahot & Pallier (2024) showed that LLMs that best predict fMRI responses to natural speech are those with the largest amount of parameters. In parallel, Tikochinski et al. (2025) and Caucheteux et al. (2023) showed that context size improved the alignment between brain and fMRI. Here, we show the effect of context size and model size to be nearly loglinearly correlated, hence pointing to diminishing returns, if not a plateau. Together, these findings clarify the specific conditions required for brain-like representations and computations to emerge. It remains unclear, however, whether context and size act directly on the alignment, or are confounded by other uncontrolled variables, such as linguistic performance. For example, in Appendix E, we find that performance is correlated with temporal alignment for specific conditions – LLM with increasing context lengths – but not others – LLMs with increasing sizes. Disentangling the causal chain that links these factors remains a major research avenue.
>
> - Antonello et al 2023  NeurIPS. Scaling laws for language encoding models in fMRI.
> - Bonnasse-Gahot & Pallier (2024), NeurIPS. fMRI predictors based on language models of increasing complexity recover brain left lateralization
> - Tikochinski et al. (2025), Nat. Comm. Incremental accumulation of linguistic context in artificial and biological neural networks
> - d’Ascoli et al. (2024), arXiv. Decoding individual words from non-invasive brain recordings across 723 participants
> - Banville et al. (2025), arXiv. Scaling laws for decoding images from brain activity
> - Caucheteux et al. (2023), Nat. Hum. Behav. Evidence of a predictive coding hierarchy in the human brain listening to speech
>
> **3. Expected results**
>
> We agree that models with many more layers are likely to have an ‘elongated processing pipeline’. However, this does not affect our analysis, because we sample the same number of layers (n=9, linearly space between 10% and 90% of the architecture) for each model.
> In addition, to clarify that our results may not be fully expected by the community, we now highlight papers that emphasize the discrepancies between LLMs and the Brain:
>
> Yet, their architecture, sensory modality, learning objective, and training regime are remarkably disjoint Romeo et al. (2018), Dupoux (2018), Evanson et al. (2025), Ghanizadeh & Dousti (2025). Furthermore, specific features like syntactic structures and semantic roles may be represented fundamentally differently (Fodor et al. (2025), Kryvosheieva & Levy (2024)). That partially converging computational paths arise despite these differences is thus all the more surprising, and highlights the necessity to clarify the computational principles that lead language processing to be partially shared between biological and artificial systems.
>
> - Romeo et al. (2018), JNeuro. Language exposure relates to structural neural connectivity in childhood
> - Dupoux (2018), Cognition. Cognitive science in the era of artificial intelligence: A roadmap for reverse-engineering the infant language-learner
> - Evanson et al. (2025), Manuscript. Emergence of language in the developing brain.
> - Ghanizadeh & Dousti (2025), arXiv. Towards data-efficient language models: A child-inspired approach to language learning
> - Fodor et al. (2025), CL, Compositionality and Sentence Meaning: Comparing Semantic Parsing and Transformers on a Challenging Sentence Similarity Dataset
> - Kryvosheieva & Levy (2024), arXiv. Controlled evaluation of syntactic knowledge in multilingual language models
>
> **4. Brain-score averaging and per sensor analysis**
>
> The brain-score is indeed the result of Pearson scores independently computed for each sensor, then averaged across the brain. We will state this averaging in the Methods section.
>
> Adding the per-sensor analyses is indeed a good idea. We now have these encoding results available and will add them to the manuscript. Note that the low spatial resolution of MEG remains challenging to explore this issue.
>
> **5. New experiments beyond LLMs: Wav2Vec2**
>
> Note that we already use two types of architectures: State Space models (can be thought of as RNNs with a linear dynamics of their hidden state) and Transformers.
>
> We now include results for speech model Wav2vec2.0, bidirectional, alongside bidirectional LLMs BERT and RoBERTa. All three show brain-scores comparable to larger, more recent and causal LLMs, but much lower temporal scores. These results are shown in the first table of the next section, alongside additional analyses.
>
> **6 New experiments: Additional models and analyses**
>
> To address reviewers’ comments, we now run five additional analyses:
>
> 1. To assess the impact of contextual directionality, we compare two bidirectional LLMs (BERT and RoBERTa) to the causal models previously presented, which reflect more closely the brain’s causal processing of language. While brain scores are comparable, the temporal scores of bidirectional models are substantially lower than those of causal LLMs.
>
> 2. We also analyse a speech model, Wav2vec2, bidirectional as well, and compare this model to the other models presented. While brain scores are comparable, the temporal scores of bidirectional Wav2vec2.0 is also substantially lower than those of causal LLMs.
>
> 3. We extend our analyses to a “large LLM”, i.e. Llama 3.3 70B. The results suggest that there is no major improvement as compared to smaller LLMs e.g. Llama 3.2 3B, hence pointing to a plateau effect of scaling laws identifiable with MEG.
>
> 4. We now added GPT2-S (137M) compared to GPT-XL (1.6B) to study scaling in this older family of models.
>
> | Model        | Brain | Temp |
> |--------------|-------|------|
> | Wav2Vec2.0   | .055  | .00  |
> | BERT         | .055  | .38  |
> | RoBERTa      | .058  | -.18 |
> | GPT2         | .054  | .20  |
> | GPT2-XL      | .054  | .87  |
> | Mamba        | .061  | .93  |
> | Mistral-7B   | .059  | .96  |
> | LLaMA-3.2-3B | .055  | .94  |
> | LLaMA-3.3-70B| .058  | .93  |
>
> 5. Finally, we now added analyses on even longer context sizes, with up to 1000 words on both Llama-3.2 and Mamba (SSM).
> The results indicate that up to 1000 words, longer contexts still lead to an increase of temporal alignment, even when the brain-score plateaus.
>
> ### Temporal Score
> | Context length | 1    | 5    | 10   | 50   | 500  | 1000 |
> |----------------|------|------|------|------|------|------|
> | Mamba-1.4b     | .65  | .61  | .82  | .91  | .91  | .93  |
> | LLaMA-3.2-3B   | .19  | .69  | .82  | .89  | .90  | .93  |
>
> ### Brain score
> | Context length | 1     | 5     | 10    | 50    | 500   | 1000 |
> |----------------|-------|-------|-------|-------|-------|-------|
> | Mamba-1.4b     | .035  | .048  | .052  | .054  | .055  | .055  |
> | LLaMA-3.2-3B   | .041  | .054  | .057  | .058  | .059  | .058  |
>
>
> Together these MEG results are consistent with the overall trends observed with fMRI (Antonello, et al (2024).
>
> We thank Reviewer tqrA again for these constructive criticisms which help strengthen this study, and better articulate how it fits the current state-of-the-art.

---

> ### Author Response · Authors · 2025-08-05
>
> Dear reviewer, in light of the added experiments and revised sections, is there any element you believe we could add or discuss to address your review?

---

> > ### Comment · Reviewer_tqrA · 2025-08-05
> >
> > No, thank you. I have raised my score to a 4. To reiterate what I stated in the final justification section, I am satisfied with the author's responses to my concerns. I am especially appreciative of the extended literature discussion involving whether these results should be expected based on prior literature, as well as the promise to include per-sensor results and added scaling results. Contingent on these changes being added to the paper, I am willing to change my score to a 4.

---

> ### Author Response · Authors · 2025-08-06
>
> We will indeed add these changes to the paper. Thank you for your comment and thorough review.

---

### Official Review · Reviewer_WHgU · 2025-07-01

**Clarity:** 3
**Significance:** 4
**Originality:** 3
**Rating:** 5
**Confidence:** 2

**Summary:**

This paper investigates the alignment between the computations performed by large language models (LLMs) and the processing observed in the human brain during language comprehension. The authors explore whether the sequence of activations in LLMs, as they process natural language, corresponds to the temporal order of brain activity measured using MEG as individuals listen to an audiobook. The study compares neural dynamics and LLM activations across different model architectures and sizes. The study reports strong temporal alignment. Early layers of the LLMs align with early brain responses, while deeper layers align with later brain activity, a pattern consistent across both transformer and recurrent architectures.

**Questions:**

No additional questions.

**Ethical Concerns:**

["NO or VERY MINOR ethics concerns only"]

**Limitations:**

yes

**Quality:**

3

**Strengths And Weaknesses:**

This is a very timely and interesting study that sheds light on high-level simialrities between human and machine information processing.

The paper is well written and clearly structured.

I appreciate the broad model comparison but would have loved to see a larger model included to see whether the trend outlined in figure 2c continues for models in the 70B parameter neighborhood.

It would have been interesting to contrast the processing order of humans and large models with older ones such as RNNs to test whether processing order is simply always like this or whether generative models are truly different and more similar to humans.

Sec 3.2 and Fig 4: It would have been interesting to study very long contexts for SSMs such as Mamba that are not, in principle, limited at all in permissible context length.

---

> ### Author Rebuttal · Authors · 2025-07-30
>
> We thank Reviewer WHgU for their thorough review. We propose to amend our manuscript to address these issues and add five additional sets of results.
>
> **1. New experiment with large language models**
>
> We now managed to implement our analysis also on model Llama 3 Instruct (70B). The obtained results don’t outperform results from smaller models, which suggests a limit to the scaling trends observed earlier. We now indicate in the discussion section:
>
> Third, our experiments show that this alignment directly depends on (i) context size and (ii) model size – although with a saturation beyond 70B parameters models. These MEG results extend previous works on scaling laws in neuro-ai (Antonello et al. (2023), Bonnasse-Gahot & Pallier (2024), Tikochinski et al. (2025), d’Ascoli et al. (2024), Banville et al. (2025), Caucheteux et al. (2023)).
> In particular, Antonello et al. (2023) and Bonnasse-Gahot & Pallier (2024) showed that LLMs that best predict fMRI responses to natural speech are those with the largest amount of parameters. In parallel, Tikochinski et al. (2025) and Caucheteux et al. (2023) showed that context size improved the alignment between brain and fMRI. Here, we show the effect of context size and model size to be nearly loglinearly correlated, hence pointing to diminishing returns, if not a plateau. Together, these findings clarify the specific conditions required for brain-like representations and computations to emerge.
> It remains unclear, however, whether context and size act directly on the alignment, or are confounded by other uncontrolled variables, such as linguistic performance. For example, in Appendix E, we find that performance is correlated with temporal alignment for specific conditions – LLM with increasing context lengths – but not others – LLMs with increasing sizes. Disentangling the causal chain that links these factors remains a major research avenue.
>
> - Antonello et al 2023  NeurIPS. Scaling laws for language encoding models in fMRI.
> - Bonnasse-Gahot & Pallier (2024), NeurIPS. fMRI predictors based on language models of increasing complexity recover brain left lateralization
> - Tikochinski et al. (2025), Nature Communications. Incremental accumulation of linguistic context in artificial and biological neural networks
> - d’Ascoli et al. (2024), arXiv. Decoding individual words from non-invasive brain recordings across 723 participants
> - Banville et al. (2025), arXiv. Scaling laws for decoding images from brain activity
> - Caucheteux et al. (2023), Nature Human Behaviour. Evidence of a predictive coding hierarchy in the human brain listening to speech
>
>
> **2. New experiment on Mamba with context size**
>
> We now extend our context size experiments to Mamba up to n=1,000 words, where temporal score keeps increasing until 0.93. As comparison, we conduct the same analysis for Llama-3.2, where temporal scores also keep increasing across increasingly long contexts. These elements suggest, with MEG, the very long context sizes do still improve the modeling of brain activity for at least some of the most recent models, State-Space and LLMs. Results are all showcased and contextualized alongside other analyses in the table of **4. New experiments**, part 5).
>
> **3. New experiments on old models and RNNs**
>
> We now extended our pipeline to older language models including GPT-2 small, BERT and Roberta. These older language models present brain-scores comparable to larger, more recent LLMs, but much lower temporal scores. These results are shown and contextualized in the first table of the next section **4. New experiments**, with additional analyses.
>
> We also analyze State-Space models (Mamba and RecurrentGemma), which can be viewed as types of RNN with a linear evolution of the hidden state at each layer. Additionally, we propose to add the analysis of a recent RNN, xLSTM (Beck et al 2024), in the final version of the manuscript.
> - Beck et al. (2024), NeurIPS. "xLSTM: Extended Long Short-Term Memory."
>
> Overall, the updated experiments do not change but extend further our original conclusion.
>
> **4. New experiments: Additional models and analyses**
>
> To address reviewers’ comments, we now run five additional analyses:
>
> 1. To assess the impact of contextual directionality, we compare two bidirectional LLMs (BERT and RoBERTa) to causal models previously showcased in the manuscript, which reflect more closely the brain’s causal processing of language. While brain scores are comparable, the temporal scores of bidirectional models are substantially lower than those of causal LLMs.
>
> 2. We also analyse a speech model, Wav2vec2, bidirectional as well, and compare this model to the other models presented. While brain scores are comparable, the temporal scores of bidirectional Wav2vec2.0 is also substantially lower than those of causal LLMs.
>
> 3. We extend our analyses to a “large LLM”, i.e. Llama 3.3 70B. The results suggest that there is no major improvement as compared to smaller LLMs e.g. Llama 3.2 3B, hence pointing to a plateau effect of scaling laws identifiable with MEG.
>
> 4. We now added GPT2-S (137M) compared to GPT-XL (1.6B) to study scaling in this older family of models.
>
> | Model          | Brain Score | Temporal Score |
> |----------------|-------------|----------------|
> | Wav2Vec2.0     | .055        | .00            |
> | BERT           | .055        | .38            |
> | RoBERTa        | .058        | -.18           |
> | GPT2           | .054        | .20            |
> | GPT2-XL        | .054        | .87            |
> | Mamba          | .061        | .93            |
> | Mistral-7B     | .059        | .96            |
> | LLaMA-3.2-3B   | .055        | .94            |
> | LLaMA-3.3-70B  | .058        | .93            |
>
>
> 5. Finally, we now added analyses on even longer context sizes, with up to 1000 words on both Llama-3.2 and Mamba (State-Space Model).
> The results indicate that up to 1000 words, longer contexts still lead to an increase of temporal alignment, even when the brain-score plateaus.
>
> ### Temporal Score
> | Context length | 1    | 5    | 10   | 50   | 500  | 1000 |
> |----------------|------|------|------|------|------|------|
> | Mamba-1.4b     | .65  | .61  | .82  | .91  | .91  | .93  |
> | LLaMA-3.2-3B   | .19  | .69  | .82  | .89  | .90  | .93  |
>
> ### Brain score
> | Context length | 1     | 5     | 10    | 50    | 500   | 1000 |
> |----------------|-------|-------|-------|-------|-------|-------|
> | Mamba-1.4b     | .035  | .048  | .052  | .054  | .055  | .055  |
> | LLaMA-3.2-3B   | .041  | .054  | .057  | .058  | .059  | .058  |
>
>
>
> We would like to thank again Reviewer WHgU for their insightful comments, and hope this further strengthens our contribution.

---

> ### Author Response · Authors · 2025-08-05
>
> Dear reviewer, in light of the added experiments and revised sections, is there any additional element you believe we could add to address your review, or something you would wish to discuss?

---

### Official Review · Reviewer_mo9D · 2025-07-03

**Clarity:** 3
**Significance:** 3
**Originality:** 3
**Rating:** 5
**Confidence:** 4

**Summary:**

This paper aims to assess whether brain---neural network representation alignment is a consequence of the two systems performing similar computations. To this end, authors make a clever experimental design wherein temporally-resolved brain signals are matched with model representations. If representations generate in a similar order, the alignment across layers will reflect that---a finding authors make.

**Questions:**

Building on one of the minor weaknesses, I would appreciate authors' thoughts on why they found model scaling to not be harmful for achieving temporal alignment. This is in contrast to prior finding on brain representational alignment. My working hypothesis would be that a good modeling of context is necessary to get good temporal alignment, which is not the target of analysis of past work, and hence scaling models eventually saturates alignment score. Alternatively, it could be the case that the temporal alignment score is too generously defined, i.e., it is trivial to achieve high scores on it. Having baselines, e.g., random or relatively less competent models (e.g., GPT-2 small), could help assess this.

**Ethical Concerns:**

["NO or VERY MINOR ethics concerns only"]

**Final Justification:**

Authors addressed all of my concerns.

**Limitations:**

Yes

**Quality:**

3

**Strengths And Weaknesses:**

**Strengths.** I really enjoyed the paper! While arguably the experiments are behavioral in nature, i.e., merely representational alignment measures are used, I found the experimental design very well motivated. The findings themselves are reasonable, though I am somewhat confused why better performing models (often the larger one) show a better temporal alignment score (see questions).

**Weaknesses.** I don't have much to note here. Adding some very minor points below.

- I think the writing can improve on points of discussion w.r.t. past work. For example, on page 4, in section 3.2, I expected authors would need to discuss working memory constraints, since a Transformer's ability to (in theory) keep track of infinitely long contexts is often used to argue against their use for neuroscientific comparisons between LLMs and human brain. Similarly, the impact of model scaling would be worth discussing at several points: authors often saw larger models are performing equally well and scoring high on temporal alignment, which stands in contrast to expectations one would have based on past work that shows scaling model does not improve brain alignment.

- Figure 6's design is somewhat awkward. Individual markers are different models and colors are model families. Connecting these markers with lines does not make sense if you're not trying to put together a trend. Arguably, that single plot in Figure 6a should be three separate plots.

---

> ### Author Rebuttal · Authors · 2025-07-30
>
> We thank Reviewer mo9D for their thorough review. We propose to amend several sections of the manuscript to address these issues and add five additional sets of results.
>
> **1. Discussion working memory**
>
> This is indeed a very good point. We will add the following paragraph:
>
>
> “Interestingly, the comparison between State Space Models (SSMs) and LLMs, offer a new perspective to investigate context-size. Indeed, transformers compute contextual representations, thanks to a non linear combination of the past context, and hence require very large memory buffers that are implausible in the brain. By contrast, SSMs can be thought of as recurrent neural networks (RNNs) with hidden states that linearly evolve over time. At each time point, they thus represent the full context with their hidden state – an approach presumably similar to the brain. Our results show that SSMs do present very large context size improvements, and thus show that it is, in fact possible, to build and maintain a long context in a single hidden state. However, further research remains necessary to evaluate the degradation of such memory in the absence of meaningful context, such as during the memorization of a random digit sequence like a phone number – a task recognized as highly constrained in human subjects (Miller et al. (1956)).”
> - Miller (1956), Psychological Review. "The magical number seven, plus or minus two: Some limits on our capacity for processing information."
>
> **2. Discussion model scaling**
>
> We agree that we insufficiently credited previous work on scaling laws in neuro-ai. We now amended the discussion as follows:
>
> Third, our experiments show that this alignment directly depends on (i) context size and (ii) model size. These MEG results extend previous works on scaling laws in neuro-ai (Antonello et al. (2023), Bonnasse-Gahot & Pallier (2024), Tikochinski et al. (2025), d’Ascoli et al. (2024), Banville et al. (2025), Caucheteux et al. (2023)).
> In particular, Antonello et al. (2023) and Bonnasse-Gahot & Pallier (2024) showed that LLMs that best predict fMRI responses to natural speech are those with the largest amount of parameters. In parallel, Tikochinski et al. (2025) and Caucheteux et al. (2023) showed that context size improved the alignment between brain and fMRI. Here, we show the effect of context size and model size to be nearly loglinearly correlated, hence pointing to diminishing returns, if not a plateau. Together, these findings clarify the specific conditions required for brain-like representations and computations to emerge. It remains unclear, however, whether context and size act directly on the alignment, or are confounded by other uncontrolled variables, such as linguistic performance. For example, in Appendix E, we find that performance is correlated with temporal alignment for specific conditions – LLM with increasing context lengths – but not others – LLMs with increasing sizes. Disentangling the causal chain that links these factors remains a major research avenue.
>
> - Antonello et al. (2023): Antonello et al 2023  NeurIPS. "Scaling laws for language encoding models in fMRI."
> - Bonnasse-Gahot & Pallier (2024): Bonnasse-Gahot & Pallier (2024), NeurIPS. "fMRI predictors based on language models of increasing complexity recover brain left lateralization."
> - Tikochinski et al. (2025): Tikochinski et al. (2025), Nature Communications. "Incremental accumulation of linguistic context in artificial and biological neural networks."
> - d’Ascoli et al. (2024): d’Ascoli et al. (2024), arXiv. "Decoding individual words from non-invasive brain recordings across 723 participants.",
> - Banville et al. (2025): Banville et al. (2025), arXiv. "Scaling laws for decoding images from brain activity."
> - Caucheteux et al. (2023): Caucheteux et al. (2023), Nature Human Behaviour. "Evidence of a predictive coding hierarchy in the human brain listening to speech."
>
>
> **3. Redesigned figure 6**
>
> We agree with this note and propose to separate the figure 6a in three plots. The line linking the markers is not crucial to the figure but its objective is indeed to highlight trends where bigger models and longer contexts induce higher brain-score and temporal-score. Separating the figure in three plots shall make these trends clearer.
>
>
> **4. Baselines as untrained or less competent model such as GPT-2**
>
> The untrained versions of the models are shown as dashed grey lines in Figure 1 of the manuscript.Their brain scores did not differ significantly from zero over time, resulting in a null temporal score.
>
> Analyzing a less competent models such as GPT-2-small is indeed a very good point. We conducted this analysis and found a brain-score of 0.054, comparable to more recent and bigger models, but a temporal score of 0.20, which is substantially lower. We further analyze and contextualize this result in the following section, alongside additional analyses.
>
>
> **5. New experiments: Additional models and analyses**
>
> To address reviewers’ comments, we now run five additional analyses:
>
> 1. To assess the impact of contextual directionality, we compare two bidirectional LLMs (BERT and RoBERTa) to the causal models previously presented, more closely reflecting the brain’s causal processing of language. While brain scores are comparable, the temporal scores of bidirectional models are substantially lower than those of causal LLMs.
>
> 2. We also analyse a speech model, Wav2vec2, bidirectional as well, and compare this model to the other models presented. While brain scores are comparable, the temporal scores of bidirectional Wav2vec2.0 is also substantially lower than those of causal LLMs.
>
> 3. We extend our analyses to a “large LLM”, i.e. Llama 3.3 70B. The results suggest that there is no major improvement as compared to smaller LLMs e.g. Llama 3.2 3B, hence pointing to a plateau effect of scaling laws identifiable with MEG.
>
> 4. We now added GPT2-S (137M) compared to GPT-XL (1.6B) to study scaling in this older family of models.
>
> | Model          | Brain Score | Temporal Score |
> |----------------|-------------|----------------|
> | Wav2Vec2.0     | .055        | .00            |
> | BERT           | .055        | .38            |
> | RoBERTa        | .058        | -.18           |
> | GPT2           | .054        | .20            |
> | GPT2-XL        | .054        | .87            |
> | Mamba          | .061        | .93            |
> | Mistral-7B     | .059        | .96            |
> | LLaMA-3.2-3B   | .055        | .94            |
> | LLaMA-3.3-70B  | .058        | .93            |
>
>
> 5. Finally, we now added analyses on even longer context sizes, with up to 1000 words on both Llama-3.2 and Mamba (State-Space Model).
> The results indicate that up to 1000 words, longer contexts still lead to an increase of temporal alignment, even when the brain-score plateaus.
>
> ### Temporal Score
> | Context length | 1    | 5    | 10   | 50   | 500  | 1000 |
> |----------------|------|------|------|------|------|------|
> | Mamba-1.4b     | .65  | .61  | .82  | .91  | .91  | .93  |
> | LLaMA-3.2-3B   | .19  | .69  | .82  | .89  | .90  | .93  |
>
> ### Brain score
> | Context length | 1     | 5     | 10    | 50    | 500   | 1000 |
> |----------------|-------|-------|-------|-------|-------|-------|
> | Mamba-1.4b     | .035  | .048  | .052  | .054  | .055  | .055  |
> | LLaMA-3.2-3B   | .041  | .054  | .057  | .058  | .059  | .058  |
>
>
> We would like to thank again Reviewer Mo9D for their insightful comments, and hope this further strengthens our contribution.

---

> > ### Comment · Reviewer_mo9D · 2025-08-02
> >
> > Thank you for the response! I'll keep my scores as is.

---

> ### Author Response · Authors · 2025-08-05
>
> Thank you for your response ! As we implemented the experiments you proposed, please let us know if there is anything else you would like to discuss, or an additional element you think we could add to increase your rating.

---

### Decision · Program_Chairs · 2025-09-17

**Decision:**

Accept (spotlight)

**Comment:**

This paper investigates similarity between human's brain and LLMs by calculated the correlation between activities of LLMs and biological brains when a long text and its corresponding audio are input. The experiments are conducted thoroughly across different type and size of LLM models, and the authors found an interesting phenomenon such that there are strong correlation between LLMs and human's brain and the temporal order of correlation is well aligned with the order of layers.

The reviewers are satisfied with the writing quality and the scientific findings of this paper. The hypothesis the authors made is well supported by the detailed and thorough experiments.